# Comparison of the Tribological Properties of the Thermal Diffusion Zinc Coating to the Classic and Heat Treated Hot-Dip Zinc Coatings

**DOI:** 10.3390/ma14071655

**Published:** 2021-03-28

**Authors:** Dariusz Jędrzejczyk, Wojciech Skotnicki

**Affiliations:** Department of Mechanical Engineering Fundamentals, University of Bielsko-Biala, Willowa 2, 43-309 Bielsko-Biała, Poland; wskotnicki@ath.bielsko.pl

**Keywords:** friction coefficient, thermal diffusion zinc galvanizing, coating hardness

## Abstract

The presented studies are focused on the wear resistance and friction coefficient changes of the thermal diffusion (TD) zinc coating deposited on steel. The aim of research was to evaluate the variation in coating properties during dry friction as a result of the method of preparation of the basis metal. The measured properties were compared to those obtained after classic hot-dip (HD) zinc galvanizing—heat treated and untreated. Thermal diffusion zinc coatings were deposited in industrial conditions (according to EN ISO 17668:2016-04) on disc-shaped samples. The results obtained during the tribological tests (T11 pin-on-disc tester) were analysed on the basis of microscopic observations (with the use of optical and scanning microscopy), EDS (point and linear) analysis and microhardness measurements. The obtained results were similar to effects observed after heat treatment of HD zinc coating. The conducted analysis proved that the method of initial steel surface preparation results in changes in the coating’s hardness, friction coefficient and wear resistance.

## 1. Introduction

Thermal diffusion (sherardizing) is a diffusion zinc coating method which is increasingly used as an alternative to hot dip zinc galvanizing, as the corrosion protection of the different small elements (fasteners, wires, bolts, screws, nails, springs, etc.). Due to important advantages (environmentally friendly process, no chromate treatment, surface ready for varnishing and vulcanization, no risk of hydrogen embrittlement), this method is constantly being developed and improved [1,2,3,4,5]. For example, in the paper [3,4], an innovative solution was proposed—the forced recirculation of the reactive atmosphere.

Structural elements are made of a wide range of materials that require various types of corrosion protection. For example, fasteners are manufactured from different metallic materials ranging from common steel, alloy steel, stainless or corrosion resistant steel to aluminium alloys and titanium [6]. Pressure to reduce the production cost means that more and more often structural elements are made of less advanced materials that guarantee only the appropriate mechanical properties. Additional functional properties, such as corrosion resistance and wear resistance, are obtained by applying appropriate coatings, whose thicknesses vary over a wide range—from nanometers [7] to several hundred micrometers [8]. To increase the wear resistance, harder and harder coatings are applied, with a hardness up to 1700 HV [9,10]. Hardness greater than 40 GPa was reported for systems based on TiN NbN, TiN VN and TiN/ZrN layers [11]. Zinc is one of the cheapest elements among those traditionally used in the production of anticorrosion coatings (Zn, Cu, Ni, Cr) [12], and moreover, processes of Zn coating deposition are very simply—they do not require large financial outlays [13]. Generally, zinc coatings applied to different elements are obtained via four methods: hot-dip galvanizing, electro-galvanizing, zinc lamella and sherardizing (thermal diffusion) [14,15,16]. In the case of some structural elements, where a very good surface representation is necessary, the requirements concern the limitation of the coating thickness. This applies among others to the bolts designed for joining structural elements [14]. In addition to corrosion resistance, an important parameter of fasteners is friction coefficient. In the case of too low a friction value, there is a potential risk of self-loosening of the joint. If the friction coefficient is too high, there is a risk of too low clamping forces, resulting in a joint failure due to incomplete tightening or complete fracture of the bolt. The requirements of a proper thread match between the bolt and nut limit the application of a hot-dip zinc galvanizing of bolts, especially with a small diameter. However, in some cases, the high-temperature (ab. 535 °C) hot-dip zinc galvanizing [17] that allows for removal of the excess zinc from the surface of bolts is applied, but such a treatment temperature can result in issues with the material of the bolts—steel tempering and losing the mechanical properties. As an alternative to electro-galvanizing, lamellar or thermal diffusion processes can be applied. According to Kania [14], the corrosion resistance of electro-galvanized bolts decreases quickly due to the small coating thickness. Moreover, the application of this method may result in contamination of the natural environment and potential hydrogen embrittlement of the steel [18,19,20]. Although lamella zinc technology increases its market share, especially in the area of fasteners, i.e. bolts, screws, nuts, springs, etc., the comparative tests of coatings (hot-dip, galvanic and lamellar) conducted in SO_2_ and NaCl environments showed that the hot-dip galvanized coating has the best anticorrosion properties [21]. Thus, because better results are reported even when zinc coatings on bolts are applied using sherardizing [14], in this paper, the hot-dip and thermal diffusion zinc coatings’ properties are compared.

The anticorrosion and tribological properties of hot-dip and sherardized zinc coatings depend on the microstructure observed on the coating cross section—Figure 1.

According to the Fe-Zn diagram [24,25,26] (Figure 1c), there are three phases occurring in the hot-dip zinc coating —Г(Fe_3_Zn_10_), δ (FeZn_10_, FeZn_7_), and ζ (FeZn_13_)—and an iron solid solution in zinc—η, which is formed on the outer surface as it is pulled out of the bath (Figure 1a). The current model [22] suggests that the sequence of the zinc coating growth is as follows: first, the Г_1_ phase is observed; next, within a few seconds, a sublayer of compact phase δ_c_ and palisade phase δ_p_ is created. There are a lot of factors that can influence the reactivity of steel (the quality/roughness of the galvanized surface [27,28,29,30,31], the kind of galvanized material [32,33,34], the alloyed elements added to the zinc bath [34,35,36,37,38,39], the metallurgical process parameters [40,41,42]) and thereby change the microstructure of the zinc coating. The coating microstructure obtained after thermal diffusion is similar to hot-dip zinc coating, but there is no η phase—Figure 1b [43,44]. However, there is also some controversy regarding the appropriate coating structure. Evans [23] claims that the outer layer is a mixture of ζ and zinc, with 7–10% iron content in the form of FeZn_7_. The second alloy layer δ contains 25% of iron in the form of Fe_11_Zn_40_. The inner layer creates an Г phase with 50% iron content. According to Jiang [1], the sherardized coatings are composed of the loose outer layer (ζ—FeZn_13_ phase) and the dense inner layer (δ—FeZn_7_ phase) with higher hardness. Konstantinov [45] analyses the two-phase structure: (Г + δ). On the other hand, Wortelen [44] stated that after sherardizing the coating structure is composed of Г, Г_1_, δ_1_ and ζ. Furthermore, an investigation conducted by Kania [14] confirmed the presence of Г_1_(Fe_11_Zn_40_) and δ_1_(FeZn_10_) phases, although according to the Fe-Zn equilibrium system, phases Г and ζ are also stabile.

The tribological properties of the zinc coating are closely correlated with its microstructure and are resultant of the properties of the phases visible in the coating cross section. The chemical formula and hardness values available in literature of the Fe-Zn intermetallic phases of the hot-dip and thermal diffusion zinc coatings are presented in Table 1.

Zinc coatings (hot-dip, galvanic, lamellar, sherardized) show a considerable differentiation of the hardness [16,48,49]—the lowest values (50 HV) are measured after the hot-dip galvanizing. Tribological properties are in direct correlation with the hardness and microstructure of the applied coating. Thus, different methods are used to improve zinc coating wear resistance by increasing its hardness. In article [50], heat treatment was applied to increase the wear resistance of hot-dip (HD) zinc coating. The coating structure formed after the conducted experiment was similar to that observed in thermal diffusion coating, i.e., there was no pure η phase and the created coating was composed of δ and ζ phases. As a result of the structure changes, the hardness of the coating increased fivefold to values close to those measured in the case of thermal diffusion (TD) coatings. 

Considering the above analysis, the aim of this paper was to compare the wear resistance of TD zinc coating to classic and heat treated HD coatings. Additionally, the experiment was focused on determination of the relation between the coating’s microstructure and measured friction coefficient value and possibilities of adjusting it to the requirements.

## 2. Materials and Methods

During the investigation, the pin-on-disc test was applied to measure the changes of the instantaneous and average values of the friction coefficient on the zinc coating cross section in the friction pair (zinc coating/steel pin). The applied test also allowed the rate of wear of the tested coatings to be determined [51,52].

The tribological investigations with application of the T11 device consisted of testing the steel pin/zinc coating couple in dry friction conditions and calculating the friction coefficient. To conduct the experiment, surfaces of the tested disc-shaped samples were subjected to friction with a Ø 4 steel rod, with a constant load of F = 9.8 N, which moved in circles on the surface of the samples at a rate of n = 45 rotations/min for a duration of 30 min. The friction coefficient was measured every 0.2 s.

Zinc coatings were deposited on the disc-shaped samples measuring 25 mm in diameter and 7 mm in thickness made of low-carbon DC01 steel (0.201% C, 0.009% P, 0.007% S, 1.01% Mn, 0.018% Al, 0.084% Si, 0.181% Cu, 0.067% Ni).

The thermal diffusion process was conducted in industrial conditions according to EN ISO 17668:2016-04 [53] in the mixture of zinc powder (99% Zn, 0.009% Pb, 0.006% Cd, <0.005% Fe, average grains size 3–4 µm) in rotary chambers that rotated at a rate of 5‒10 turns per minute, at a temperature of 400 °C, for a period of 4 h. The disc-shaped samples’ surface before TD zinc deposition was prepared in different ways—grinded with sandpaper with gradations 30 (TD30), 60 (TD60), 120 (TD120), 240(TD240), sandblasted (SB) and turned (T). Samples used for comparison were hot-dip galvanized (marked as untreated—HDUT) according to EN ISO 10684 [54]—a process of etching in 12% HCl, fluxing and dipping in a Zn bath with Al (0.002%), Bi (0.055%) and Ni (0.058%), at a temperature of 460 °C within 1.5 min, followed by cooling in water. In addition, the heat treated HD galvanized samples were used for comparison (HDHT—temperature T = 430 °C, τ = 7 min [50]). The following parameters were analysed during investigations: the wear resistance —disc-shaped samples weight loss, the friction coefficient (T11 pin-on-disc tester); the microstructure of the zinc coating structure and steel using an Axiovert 100 A optical microscope (Zeiss Group, Oberkochen, Germany) and an EVO 25 MA Zeiss scanning electron microscope with an EDS attachment (Zeiss Group, Oberkochen, Germany); and microhardness changes in the cross section of both the coating and the subsurface layer of steel (Vicker’s HV 0.02, Mitutoyo Micro-Vickers HM-210A device 810-401 D, Mitutoyo Corporation, Kanagawa, Japan). Additionally, the surface roughness was measured using an optical Phase View ZeeScan system (Phase View, Paris, France). The test samples were carefully prepared in order to avoid the overheating and spalling during cutting (hand cutting, hot embedded, grinded and polished).

## 3. Results and Discussion

### 3.1. Metallographic Observations and Microhardness Distribution

The zinc coating thickness measured during the microscopic observations was verified through measurements in a wider range with the use of the magnetic induction method—an electronic PosiTector 6000 tester (DeFelsko Corporation, Ogdensburg, NY, USA). The thickness of the coating on the disc-shaped samples, after hot dip and thermal diffusion galvanizing, was in the range of 45–55 µm.

The TD zinc coating morphology presented in Figure 2 is in accordance with the literature data [22,23,24]. It is very difficult to distinguish between the different phases in the coating using SEM observation—Figure 2a. Only the linear and point EDS analysis shows the existence of two areas—Figure 2b, Table 2. The outer layer has a higher Zn content than the inner one adjoined to the basis metal—steel. Taking into account that the first point of the EDS analysis was several microns away from the outer surface and the trend in the course of the Zn linear analysis, which was clearly downward near the surface, it may be assumed that a δ phase (FeZn_7_ or FeZn_10_) was present in the outer layer [16,45,46]. The chemical composition of the zone close to the basis metal suggests that a Г_1_ phase is located in this area [14,16,46].

The tribological properties (weight loss, friction coefficient) of the TD coating were compared to analogical data concerning HD and heat treated HD coatings. The classical HD coating structure is composed of four phases, whereas in the structure after heat treatment, three phases are visible (η phase is missing). The typical microstructure of a tested HD zinc coating formed by phases η, ζ, δ and Г1 [8], is shown in Figure 3a.

Most of the data [22,55] confirm that there are only three phases in a HD zinc coating after heat treatment: Г (23.5–28.0 wt% Fe), δ (7.0–11.5 wt% Fe) and ζ (6.0–6.2 wt% Fe)—Figure 4 [46,56]. During the heat treatment, the δ and Γ phases grow at the expense of the ζ phase [57], and at higher temperatures, the ζ layer disappears and in its place the δ phase grows reaching to the surface of the coating [24]. The conditions for the growth of individual phases here are similar to the TD process but the Zn amount in the coating is constant and the coating thickness is stable during the treatment [50].

Microscopic examinations (both optical—Figure 5—and scanning microscope—Figure 2a and Figure 4a) showed that the outer sublayer of TD coating was slightly cracked and porous to a depth of 10 micrometres, whereas there were no discontinuities, porosities, cracks or surface degradation visible as a result of the conducted heat treatment of the hot-dip zinc coating.

The profiles of the hardness changes on the cross sections of zinc coating and subsurface steel area are presented in Figure 6. Analysis of the obtained results showed that there are no essential differences between measured hardness values of TD coatings deposited on the steel surface with various surface conditions. The hardness values in the coatings’ outer layer were in the range 370–385 HV 0.02, while the values measured in the inner layer were in the range 325–345 HV 0.02. The downward trend in hardness changes was observed over the entire cross section of the coatings. The highest average hardness values were measured in the outer coating layer for the surface grinded using sandpaper with gradation 30 and turned (385 and 383 HV 0.02). The lowest hardness values over the entire cross section of the coating were measured for the steel surface grinded with sandpaper with gradation 240 and sandblasted. 

The presented hardness changes are due to the changes in the microstructure (Г_1_+δ–suggested by results of EDS analysis), caused by diffusion of iron from the steel surface into the coating (Figure 2, Table 2). The hardness values measured by Pokorny [25] show that the δ phase is generally about 10% harder than the Г phase; the obtained hardness values of the δ phase were even in the range 330 to 460 HV. According to data [16,46], the δ phase in TD coating is harder by about 15% than the Г phase. In the analysed results of the current study, the difference is within the range 10–12%, but the coating micro-cracks may affect the measured hardness values and can have a decisive importance here.

For the compared samples (hot dip galvanized: untreated (UT) and heat treated in 430 °C (HT430 °C)), results are consistent with the literature data concerning the individual phases that were obtained (Figure 6b) [50,58]. The outer area of the TD coating is 300 HV, which is 0.02 harder than the analogic layer of the HD UT sample and 90 HV 0.02 harder than the HD HT430 °C coating.

The structure observed in the compared coatings (Figure 2, Figure 3, Figure 4 and Figure 5) deposited on the tested disc-shaped samples corresponds well with the measured microhardness distribution in the coatings and the cross section of subsurface steel layers. The steel area close to the zinc/steel surface is slightly softer (in comparison to the UT sample), as a result of overheating. At a distance of 75 µm from the steel surface, the measured average hardness values were as follows: 195 (for HD UT samples), 165 (TD samples) and 175 HV 0.02 (HD HT430 °C samples).

### 3.2. Friction Coefficient Measurements

The coating showed higher abrasion resistance with increasing of the initial steel surface roughness, which was reflected by a reduction in weight loss (Table 3, Figure 7). The difference was particularly significant in the case of TD30 and TD60 samples. The difference in weight loss between the heat treated HD and TD samples was very small (max. 0.004 g), whereas the weight loss of HD UT zinc galvanized samples was 4‒7 times higher with reference to both TD and HD HT samples.

The base steel surface roughness exerts influence also on formed thickness of the zinc coatings. Coating thickness increases with the increase of steel surface roughness as a result of the higher reactivity of the basis metal (Figure 7b). The higher increment of the coating thickness was observed when comparing TD240 and TDSB coatings (the biggest difference in roughness).

The comparison of the investigated coating’s appearance observed after the “pin-on-disc” test is presented in Figure 8. The assessment of the external appearance of the TD coatings in “macro” scale revealed that there are no visible cracks and discontinuities on the surface of the samples (before the friction test). The presence of the cracks in the upper part of the coating was confirmed only via microscopic observations (Figure 2a and Figure 5a), but occurrence of the transverse cracks is characteristic of the intermetallic Fe-Zn phases [14]. The coatings’ colour (dark grey) is similar to that seen on the HD HT samples—Figure 8a,d. As a result of the friction test, the regular groove was rubbed over the entire circuit of the tested coatings. The friction products formed on the coating surface during the test had a “coarse powder" shape with granularity depending on steel surface development (higher roughness-coarse grains). The HDUT coating was much lighter and the rubbed away particles were shaped like flakes, up to 0.4 mm long (Figure 8c). This confirmed the higher plasticity of this coating with comparison to TD and HDHT coatings.

Figure 9 shows the course of changes in the instantaneous values of the friction coefficient of the TD zinc coatings. In the process of friction, three main stages can be distinguished. In the initial period of cooperation, the friction coefficient increased rapidly and, after forming a contact, dropped down to the value 0.18–0.24. In the second stage, its value gradually increased and finally stabilized (third stage) in the range of 0.27–0.43. The average values of obtained friction coefficient were in the range 0.20–0.39 (Table 3). The measured friction coefficient values correspond well with the microhardness and weight loss trends of changes presented in Figure 6 and Figure 7. The TD coating layer was composed of a mixture of δ and Г phases in different proportions. It is probable that increasing the degree of development of the base steel surface and its roughness causes both an increase in the thickness and the hardness of the coating. The above changes may be caused by the increase of the steel reactivity and in consequence the extended range of occurrence of the harder δ phase. Therefore, the friction coefficient of the coatings TDT, TD30 and TDSB shows lower values (0.18–0.33) for an extended period of time than the coefficient determined for the coatings TD120, 240 and SB, which constantly shows a strong upward trend and stabilizes only after about 1000 s, at the level 0.42–0.44.

In the case of the reference samples (HDUT and HDHT), the tendency to change the hardness along the cross section of the coating is quite different (in comparison to TD coating—Figure 6b)—the hardness increases from the outer surface into the coating. In the HD UT coating the η phase was relatively soft (about 55 HV 0.02) and probably was partially removed during the first stage of friction (grinding-in of the pin-and-disc sample contact). The appearance of a mixture of the ζ and δ phases resulted in a reduction in the coefficient of friction from the value of 0.29 (η) to approximately 0.25 (ζ + δ)—Figure 10. After the HT (430 °C), the subsurface coating layer was composed of a mixture of η and ζ phases [53]. Therefore, the value of the friction coefficient was lower—0.28—and decreased slightly to 0.21, as the layers closer to the steel surface were rubbed.

## 4. Conclusions

(1)The method of the base steel surface preparation affects the friction coefficient value, thickness and wear resistance of the TD zinc coating.(2)In the applied test conditions, the value of the friction coefficient of the TD coating varied within the range 0.20–0.39, with a coating thickness of 44.5 to 47.5 µm, respectively.(3)The measured friction coefficient values correspond well with the microhardness profile determined on the cross section of the TD coating and the weight loss trend of changes obtained during the “pin-on-disc” test. With increasing of the coating’s hardness, both the TD coating’s coefficient of friction and the weight loss are reduced.(4)The lower values of the friction coefficient were measured for samples with higher roughness of the base steel surface. The observed changes may be caused by the increase of the steel reactivity and in consequence extending the range of occurrence of the harder δ phase.(5)The changes in properties of compared coatings are due to the differentiation in the microstructure (verified by results of EDS analysis), caused by specific growth or diffusion conditions during individual coating formation. The TD coating was composed of δ (outer) and Г (inner) phases. The microstructure of a tested HD zinc coating was formed by phases η, ζ, δ and Г1, whereas in a HD zinc coating after heat treatment, only three phases occurred: ζ, δ and Г.(6)The TD coating (δ+Г) showed higher abrasion resistance (in comparison to HD UT coating—η+ζ+δ), which was expressed in a reduction in weight loss measured during the tribological test. In the conducted test the HD zinc coating weight loss was four times greater.(7)The abrasion resistance of the TD zinc coating (δ+Г) is similar to the HD HT coating (ζ+δ+Г)—the measured difference in the weight loss was a maximum of 0.004 g.(8)The hardness of the TD zinc coating reached the values of 325–385 HV 0.02 and was greater by 40–90 HV 0.02 than the values obtained for the HD HT zinc coating.(9)With the use of the proper method of the base steel surface preparation, it is possible to improve the tribological properties of the thermal diffusion zinc coating, decrease its wear and to adjust/change the coefficient of friction according to the requirements within the range of 0.20–0.39.

## Figures and Tables

**Figure 1 materials-14-01655-f001:**
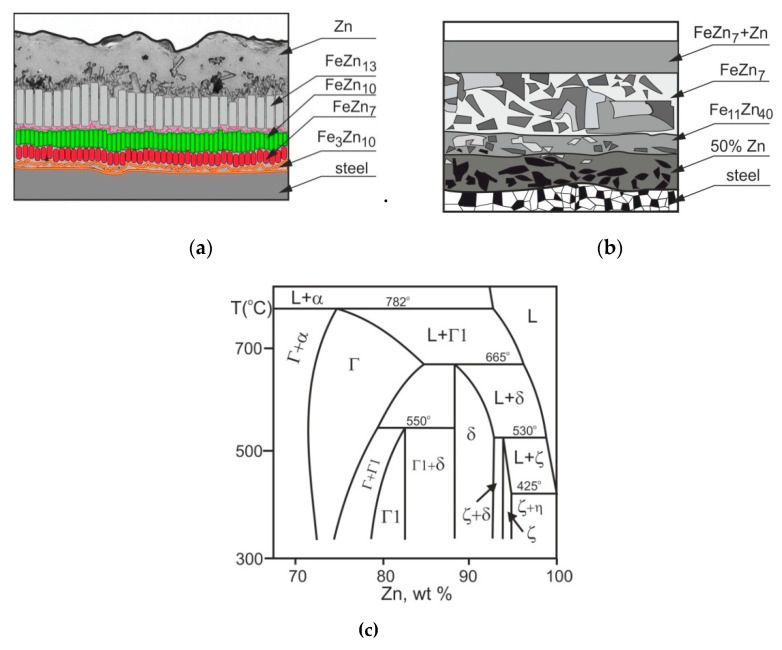
The microstructure of the zinc galvanizing coatings in relation to the Fe-Zn system; (**a**)—hot-dip [22], (**b**)—thermal diffusion [23], (**c**)—Fe-Zn equilibrium system [24,25].

**Figure 2 materials-14-01655-f002:**
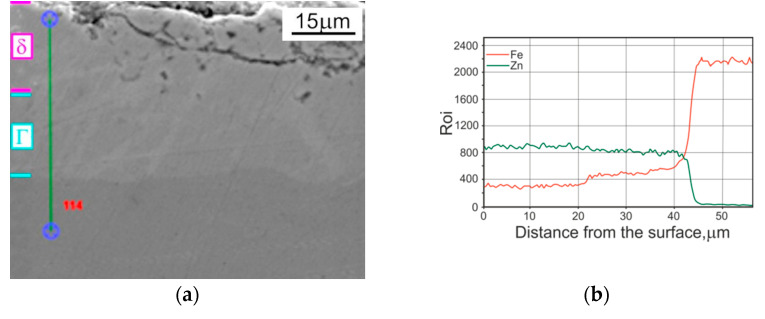
The microstructure observed at the cross section of the TD zinc coating deposited on disc-shaped samples: (**a**) SEM; (**b**) linear EDS analysis.

**Figure 3 materials-14-01655-f003:**
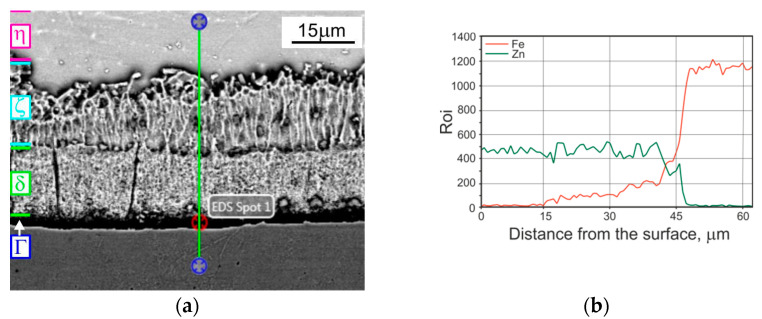
The microstructure observed at the cross section of the tested HD zinc coating deposited on disc-shaped samples: (**a**) SEM; (**b**) linear EDS analysis.

**Figure 4 materials-14-01655-f004:**
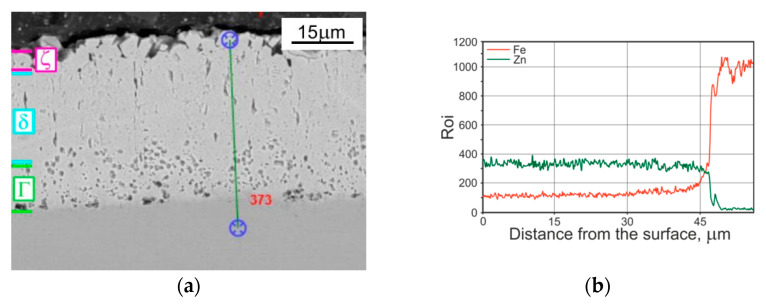
The microstructure observed at the cross section of the heat treated HD zinc coating deposited on disc-shaped samples: (**a**) SEM; (**b**) linear EDS analysis.

**Figure 5 materials-14-01655-f005:**
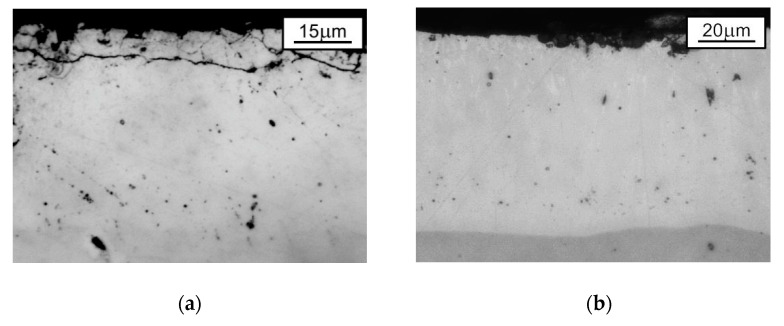
The microstructure (optical microscope images) observed at the cross section of the tested zinc coatings deposited on disc-shaped samples: (**a**) TD coating; (**b**) HDHT coating.

**Figure 6 materials-14-01655-f006:**
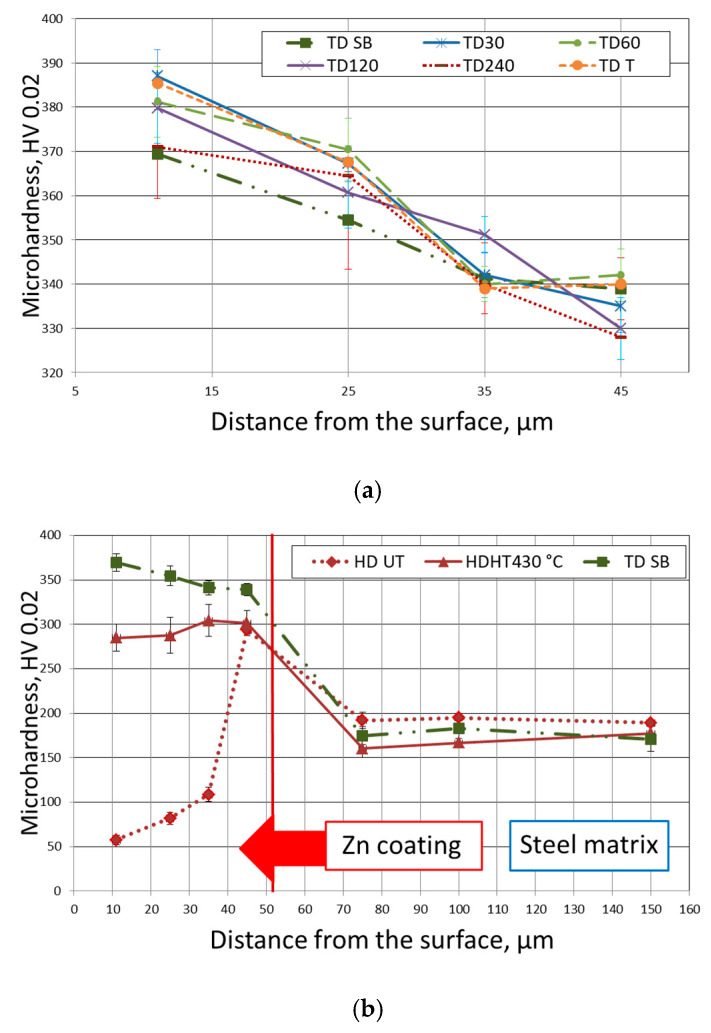
Microhardness measured at the cross section of the TD zinc coating deposited on disc-shaped steel samples (**a**) and comparison of obtained results to the microhardness measured after HD galvanizing—untreated (HD UT)—and after heat treatment at 430 ° (HD HT 430 °C) (**b**).

**Figure 7 materials-14-01655-f007:**
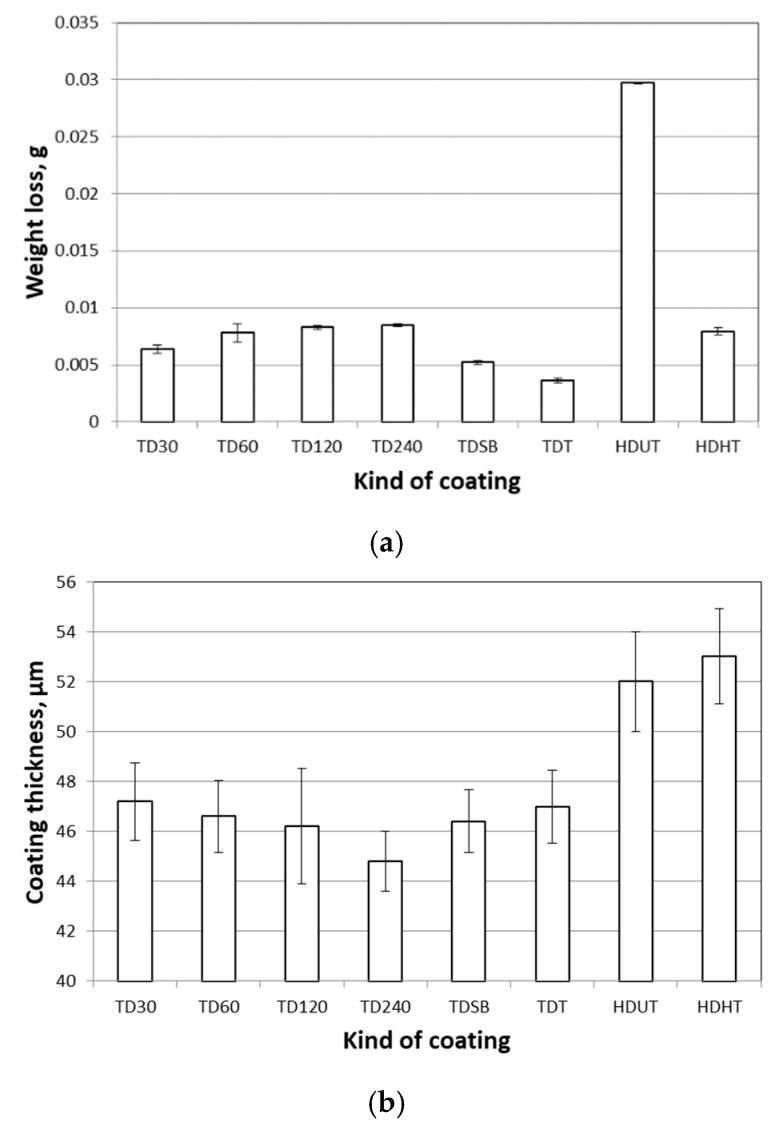
The disc-shaped samples’ weight loss after the pin-on-disc friction test (**a**) and comparison of the thickness of the TD and HD zinc coatings (**b**).

**Figure 8 materials-14-01655-f008:**
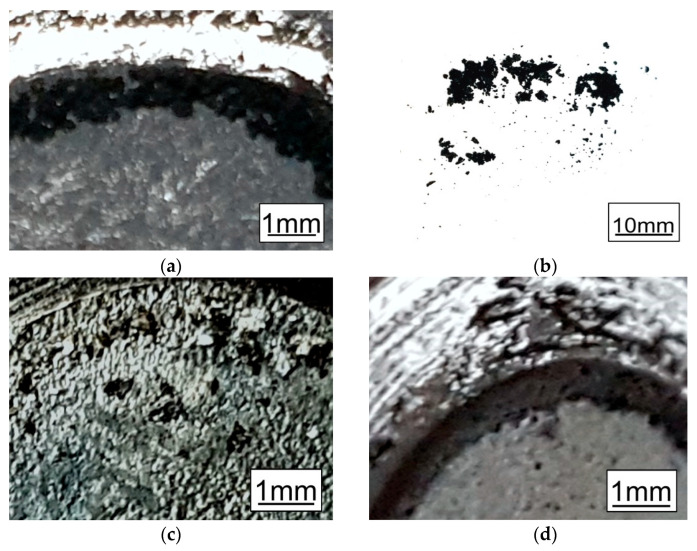
The appearance of the coatings’ outside surface and friction products observed after the pin-on-disc friction test: (**a**), (**b**) sample TD30; (**c**) sample HDUT; (**d**) sample HD HT at 430 °C.

**Figure 9 materials-14-01655-f009:**
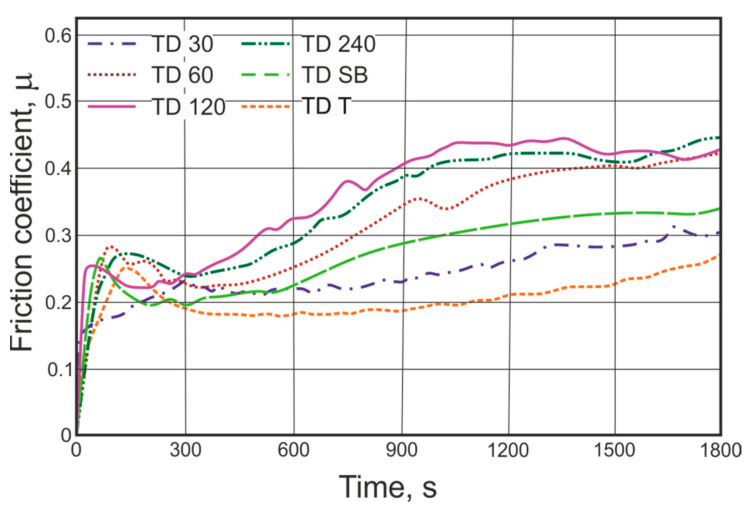
The friction coefficient values registered during pin-on-disc testing of TD zinc coatings.

**Figure 10 materials-14-01655-f010:**
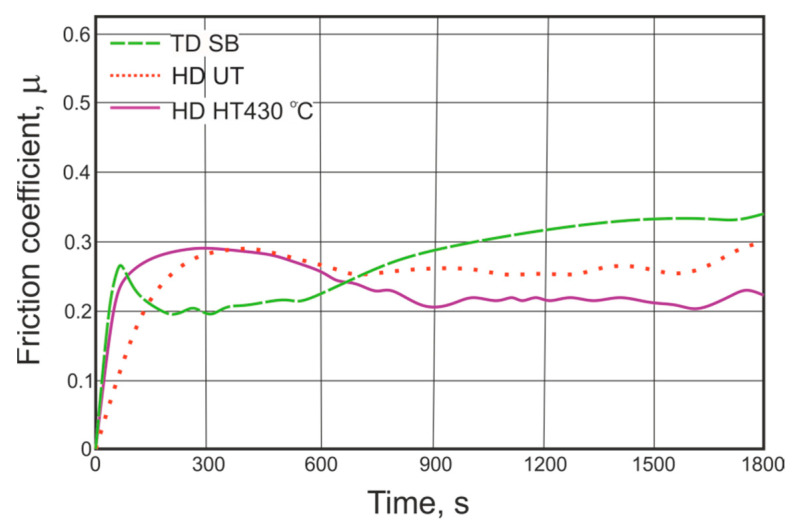
The comparison of the friction coefficient values registered during testing of the TDSB and HDUT, HDHT coatings.

**Table 1 materials-14-01655-t001:** Chemical formula and hardness of Fe-Zn intermetallic phases.

Phase	Iron Content, wt. %	Hardness	
		HD Coating	TD Coating
η–Zn	0.03 [46]	52 HV [16] 70 HB [23]	n.d. n.d.
ζ–FeZn_13_	6 [43]; 6.17 [46]; 5–6 [16]; 5.9–7.1[47]	208 HV [16] 220 HB [23]	n.d. n.d.
δ–FeZn_10_	7–11.5 [16]; 7.87 [46]	358HV [16]	n.d.
δ–(FeZn_11_–FeZn_6.67_)	8.1–13.2 [47]	n.d.	n.d.
δ–FeZn_7_	7–10 [43]; 10.87 [46]	270 HB [23]	300 HB [23]
Г1–Fe_5_Zn_21_	17–19.6 [16]; 16.90 [46]	505 HV [16]	n.d.
Г1–Fe_11_Zn_40_	19.02 [46]	n.d.	350 HB [23]
Г–Fe_3_Zn_10_	23.5–28 [16]; 20.40 [46]	326 HV [23]	n.d.
Г–(Fe_5_Zn_21_–Fe_4_Zn_9_)	18–31 [47]	n.d.	600 HB [23]

n.d.—not determined.

**Table 2 materials-14-01655-t002:** The results of EDS analysis of the TD coating deposited on disc-shaped sample.

**Element**	**Distance from the Coating Surface, µm**							
	5	10	15	20	25	30	35	45
	**Content of the Element, Wt%**							
**Fe**	11.07	11.10	10.96	13.20	18.03	20.21	22.45	91.95
**Zn**	88.93	88.90	89.04	86.80	81.97	79.26	77.55	1.13

**Table 3 materials-14-01655-t003:** The roughness and friction coefficient measured on the surface of the disc samples.

**Roughness (S_a_), µm**							
TD30	TD60	TD120	TD240	SB	T	HD	HDHT
**Before Galvanizing**							
6.22	5.66	4.59	4.19	6.80	8.38	3.63	2.29
**After Galvanizing**							
2.50	2.81	2.60	2.44	2.84	3.35	2.94	3.43
**Average Friction Coefficient Value of Zinc Coating, µ**							
0.25	0.36	0.39	0.38	0.28	0.20	0.27	0.21

## Data Availability

Data is contained within the article.

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
