# Peer review of "Comparison of the Tribological Properties of the Thermal Diffusion Zinc Coating to the Classic and Heat Treated Hot-Dip Zinc Coatings"

_materials, 2021, doi:10.3390/ma14071655_

Round 1
Reviewer 1 Report
In the submitted manuscript, the author tried to illustrate the effect of the method of the base steel surface preparation on the thermal diffusion (TD) zinc coating properties .The author also tried to compare the tribological properties of the thermal diffusion zinc coating to the classic and heat treated hot-dip zinc coatings by analyzing the change of friction coefficient value, thickness and wears resistance of the coating.However, there are still some contents that need to be checked and modified in the manuscript.
- In the introduction, there are not enough references to introduce the thermal diffusion (TD) zinc coating, and then further compare of TD zinc coating and hot-dip galvanized coating.
- In lines 173 to 176, can the author provide the optical microscope images of the outer sublayer of two kinds of zinc coatings and further analyze and explain the results of the microscopic examinations?
- Does the author have any experimental results to support this sentence between lines 198 and 199?
- Does the author have the result of the rate of wear of the tested coatings mentioned?
- The obtained other results of TD zinc coating are similar to effects observed after HDHT zinc coating, but can the author explain the difference in coating thickness?
- Does TD zinc coating have more advantages and applications?
- Is there any difference in corrosion resistance of zinc coatings obtained by different methods?
Reviewer 2 Report
The manuscript requires significant corrections:
- In the introduction, it makes sense to add a diagram of the Fe-Zn state, so that you can clearly see the phases formed in the coating during the diffusion saturation of steel with zinc;
- The section "Methods and materials" should be substantially revised. Namely, to describe in detail the method of coating with an indication of the granulometric composition of the powder, the chemical composition of the steel sample on which the coating is applied. In the same section, you need to transfer the description of the method for determining the coefficient of friction and be sure to add the method of wear resistance;
- in the figures with microstructures 2a, 3a, and 4a, the phase composition should be indicated, not the chemical composition, similar to Figure 1;
- The conclusions should clearly reflect the relationship between the coefficient of friction and wear resistance with the phase compositions of the coatings under study.
Round 2
Reviewer 2 Report
In this revision, the manuscript has become better.